# Description and Classification of Training Drills, Based on Biomechanical and Physiological Load, in Elite Basketball

**DOI:** 10.3390/s25010262

**Published:** 2025-01-05

**Authors:** Carlos Sosa, Enrique Alonso-Pérez-Chao, Carlos Ribas, Xavier Schelling, Alberto Lorenzo

**Affiliations:** 1Facultad de Ciencias de la Actividad Física y del Deporte-INEF, Universidad Politécnica de Madrid, 28040 Madrid, Spain; sosamarin.carlos@gmail.com; 2Faculty of Sports Sciences, Universidad Alfonso X el Sabio, 28691 Villanueva de la Cañada, Spain; eperezch@uax.es; 3Faculty of Sports Science, Universidad Europea de Madrid, 28001 Villaviciosa de Odón, Spain; enrique.alonso@universidadeuropea.es; 4Institute for Health and Sport (iHeS), Victoria University, Melbourne 8801, Australia; xschelling@gmail.com

**Keywords:** team sports, local positioning system, load monitoring, game demands, training

## Abstract

The aim of this study was to understand and describe the physiological and biomechanical demands of various tasks used in basketball training and, subsequently, to provide a practical application of these tasks in a typical training week. Twelve basketball players had their external load variables monitored across 179 training sessions (2896 samples) using local positioning system technology. These variables included total distance covered, distance covered at various intensity levels, accelerations, decelerations, PlayerLoad™, and explosive efforts. The analysis revealed significant differences in both physiological and biomechanical loads across various drills. Specifically, tasks with more space and fewer defenders, such as 3v0 full court, impose higher physiological loads compared to tasks with less space and more defenders, like 5v5 full court. The difference in physiological load between these tasks was statistically significant (*p* < 0.05) with a moderate effect size (ES: −0.60, 95% CI: [−0.99, −0.22]). In terms of biomechanical load, drills with increased defensive pressure, such as 5v5 full court, exhibited significantly higher values compared to less specific drills, such as 5v0 full court, with a very large effect size (ES: 1.37, 95% CI: [1.04, 1.70], *p* < 0.01). Additionally, comparisons between 5v5 full court and 3v0 full court for biomechanical load produced a very large effect size (ES: 1.67, 95% CI: [1.37, 1.97], *p* < 0.01), indicating a substantial difference in load demands. The results indicate that tasks with more space and fewer defenders impose higher physiological loads, while those with less space and more defenders increase the biomechanical load. For training design, it is recommended to schedule tasks with a higher biomechanical load at the beginning of the session and those with a physiological orientation toward the end. Understanding the distinct demands of different drills can help coaches structure training sessions more effectively to optimize player load and performance development throughout the week.

## 1. Introduction

In recent years, extensive research in team sports, particularly basketball, has focused on characterizing the game’s physical demands [1,2], identifying the most demanding passages of play [3,4], understanding the effects of training modifications during practice sessions, and exploring injury mechanisms [5]. For instance, load quantification in basketball has been employed to analyze average and peak physical demands in competition [2,6], compare practice and competition loads [7,8], observe the evolution of training loads and performance throughout a season [9], assess the impact of targeted training programs on physical performance [10], and profiling players based on age [8,11], gender [12], position [13,14], and competitive level [15,16].

This body of research improves our understanding of basketball from a physical demands’ perspective, highlighting its nature as an intermittent sport characterized by alternating offensive and defensive actions [17]. The game features frequent changes in movement type and intensity [18], with high-intensity periods interspersed with medium- and low-intensity intervals, where actions often occur unpredictably [19]. Consequently, basketball matches exhibit an irregular alternation of aerobic and anaerobic physical demands [20] that impose various neuromuscular and metabolic challenges throughout the match. Biomechanical, physiological, technical, and tactical demands in basketball create significant variability in movements and intensities, contributing to its highly intermittent nature [15,21,22].

Despite the abundance of available indicators, significant confusion and inconsistency remain in their application and integration into training processes. The current basketball literature is often more descriptive than practical and offers limited information on physiological or biomechanical stressors. As Russell et al. (2020) noted in their systematic review, there is a clear disconnect between applied practices and methodological frameworks. Given the limitations of existing studies, drawing definitive conclusions about the true physical demands of basketball is not yet possible. Most research to date (e.g., [4,5,23]) has analyzed the loads experienced by basketball players using the traditional distinction between external and internal loading. However, several studies have highlighted the challenges and critical importance of distinguishing between physiological and biomechanical load–response pathways in team sports, including basketball [23].

Monitoring physiological and biomechanical load adaptation in team sports, particularly basketball, has gained increasing recognition and research interest. Although physiological loads primarily focus on the work–energy relationship as players move around the court, biomechanical loads refer to the external forces exerted on the players through their movements. Biomechanical loads refer to the external forces exerted on players through their movements, including the impact of gravity, ground reaction forces, resistance from equipment, and interactions with opponents. These loads include the stresses placed on muscles, tendons, bones, and joints during physical activities, which influences both performance and the risk of injury [24]. It is well established that biomechanical and neuromuscular factors play a critical role in horizontal deceleration, a key component of sports that involve multidirectional movements, such as basketball. The unique ground reaction force profile during horizontal deceleration, characterized by high-impact peak forces and loading rates, can increase susceptibility to excessive forces and the risk of injury if the limbs are unable to tolerate these forces [25]. Various metrics are available to quantify the biomechanical loads experienced by the body, its structures, and individual tissues. However, the challenge lies in measuring these loads accurately, both in and out of laboratory settings [26]. The biomechanical load, defined as the external forces acting on an athlete’s body, can be monitored using Inertial Measurement Units (IMUs). These devices utilize inertial movement analysis (IMA) to capture biomechanical load variables such as player load or jumps. IMUs are widely used to monitor adaptations to training and their relationship with game performance.

For example, in football, Mandorino et al. [27] employed machine learning techniques to develop a novel locomotor efficiency index (LEI) to assess the neuromuscular fitness of players. Subsequently, Mandorino et al. [28] analyzed the effects of different training periodization strategies on the neuromuscular state of football players. In basketball, a study quantifying workload during basketball-specific drills using microtechnology revealed that full-court 3v3 and 5v5 drills imposed the highest physical demands compared to traditional balanced basketball drills such as 2v2 and 4v4. Metrics such as acceleration load per minute (AL·min^−1^) were used to assess workload, demonstrating that the drill format significantly influences the biomechanical load experienced by players [29].

More recently, Olthof et al. (2021) studied the statistical relationships between biomechanical loads in training and game performance. Their findings indicated that training loads significantly affected match loads in subsequent games. In particular, increasing training loads two days before a match led to higher expected match loads, suggesting that biomechanical loads are strong predictors of game performance.

What is clear so far is that manipulating certain variables, such as the number of players involved in the use of full-court versus half-court drills, results in significantly different player loads [2,30]. Considering these findings, the biomechanical load of various basketball exercises must be considered, with the overarching goal of training being to prepare players for competition. Training sessions should be designed to reflect this goal by observing the distinct loads induced by each exercise, not only in terms of internal and external load, but also in relation to physiological and biomechanical load. By understanding the specific physiological and biomechanical demands, internal or external, of each task assigned to players, the designs of the training sessions can be optimized, ultimately leading to improved performance in competition. Addressing the gap between theoretical knowledge and practical application is crucial because it ensures that research findings are not confined to academic settings but are effectively translated into real-world practice. In the context of basketball training, presenting actionable strategies enables coaches and practitioners to directly apply evidence-based insights to their training designs. This not only enhances the relevance and utility of the research but also helps optimize training effectiveness, ensuring that players are better prepared for the demands of competition.

To our knowledge, there is limited research that specifically differentiates between the types of loads (physiological and/or biomechanical) for each task in basketball. Furthermore, even fewer studies go beyond describing the load and offer practical applications of this knowledge for improved training design. Thus, the aim of this article is twofold; first, to identify and describe the physiological and biomechanical load associated with various tasks used in basketball training and, second, to propose a practical application of these tasks within the framework of a typical training week.

## 2. Materials and Methods

### 2.1. Sample

Elite male basketball players [31] from the same team competing in the highest regional division of an U18 Spanish basketball competition were included in this study (n = 18, mean ± standard deviation [SD]: age 16.9 ± 0.8 years, height 196.6 ± 9.4 cm, body mass: 91.7 ± 8.2 kg). Monitoring took place during 179 training sessions.

Data collection was carried out at the same facility for two consecutive seasons (2018–2019 and 2019–2020). To be included in the study, players had to complete a minimum of 50% of training sessions (n = 90/179) during both seasons; those who did not meet this criterion were excluded from the analysis. Additionally, data from players who did not complete at least 80% of the total duration of a specific training session were excluded from that session’s data pool but remained in the overarching study.

After applying the exclusion criteria, six participants who entered the study were excluded from the analysis. Consequently, 2896 training data samples from a collective of 12 participants were subjected to analysis. This study was conducted in accordance to the Declaration of Helsinki [32].

### 2.2. Procedures

This observational investigation was conducted over a 2-year period throughout the 2019–2020 and 2020–2021 seasons. Each player wore a device (Vector S7; Catapult Sports, Melbourne, Australia) in a specially designed pocket within a vest, placed on the upper thoracic spine between the scapulae. The devices contained an accelerometer (±16 g, 100 Hz), magnetometer (±4900 µT, 100 Hz), gyroscope (up to 2000 deg/s, 100 Hz), and LPS. The ClearSky LPS (ClearSky S7, 10 Hz, firmware version 5.6.; Catapult Sports, Melbourne, Australia) is an ultra-wide band, 4 GHz transmitting system equipped with 24 anchors positioned around the perimeter of a basketball stadium that was used to collect LPS data. The technology used in this study has been supported as valid in measuring distance [33,34,35,36], speed, accelerations, decelerations [33,34], and Player Load^TM^ [37], while similar LPS technology has been shown to be reliable (coefficient of variation (CV) < 5%) in measuring distance and speed variables [36]. All players were familiar with monitoring technology, having worn the devices during training and games in the previous season. Each device was turned on ~20–40 min before the warm-up that preceded each game. The players wore the same device throughout the study period to avoid variation between devices in the output of external load data [38].

Activity editing occurred during and after the session. To minimize significant interobserver variability, the editing process for all activities was carried out consistently by the same individual. During training sessions, duration was defined as the time in minutes that a player actively participated in training, excluding the intervals between exercises, hydration breaks, or instances when a player, during a task, was not actively involved. A player was considered inactive during a task if they were off the court and did not participate (e.g., in a 5v5, where a player awaits off-court to substitute for a teammate). After completing data collection, Catapult Sports Openfield cloud software (version 1.22.0) was used to extract data from each player for each training session, segmented by task. Subsequently, following the predefined exclusion criteria, the collected data were exported into a Microsoft Excel spreadsheet (version 16.0, Microsoft Corporation, Redmond, WA) for further analysis. The drills were classified according to their specificity from 0 to 5, following the classification by [29]. Activities at level 0–1 were those carried out outside the basketball court and unrelated to basketball practice (e.g., cycling), while level 5 represented an official basketball game (Table 1).

### 2.3. Physical Variables

The selected physical parameters were classified into two types (physiological and biomechanical variables) [23,39]. Each variable was extracted and represented as a relative value, indicating the rate of accumulation of that parameter per minute.

#### 2.3.1. Physiological Variables

The following 5 variables were considered physiological: distance (m) per minute covered (TD) and distance (m) per minute covered in different intensity zones including: standing–walking (S-W) = <7 km·h^−1^; jogging (JOG) = 7–14 km·h^−1^; running (RUN) = 14.01–18 km·h^−1^; and high-speed running (HSR) >18 km·h^−1^, as previously used in basketball research [15].

To classify tasks according to the orientation of the physiological load, a two-step cluster analysis was performed (average silhouette = 0.5) using physiological parameters: total distance per minute, and distance per minute at different thresholds (Table 2). Tasks were grouped into four categories: low physiological load, medium physiological load, high physiological load, and very high physiological load. Each category was assigned a numerical value, with 1 representing low physiological load, 2 for medium physiological load, 3 for high physiological load, and 4 for very high physiological load.

To determine the physiological load of each task, an average was calculated for each task based on the numerical value of the cluster load ranging from 1 to 4. For instance, if 100 official match records were distributed with 50 in cluster number 4 and 50 in cluster number 3, the average of the 100 records would be a value of 3.5, indicating a physiological load of 3.5.

#### 2.3.2. Biomechanical Variables

The following 5 variables were considered biomechanical: jumps per minute (JUMPS) > 20 cm, accelerations per minute (ACC) (count) performed > 2 m·s^−2^ (dwell time: 0.3 s), decelerations per minute (DEC) (count) performed > −2 m·s^−2^ (dwell time: 0.3 s), PlayerLoad™ per minute (PL) (arbitrary units [AU]), and explosive efforts per minute (EE). These dwell times were chosen given values between 0.3 and 0.4 s have been identified as the most readily used in basketball settings [40,41,42].

PL was calculated as the square root of the sum of the instantaneous rate of change in acceleration in the three movement planes (x-, y-, and z-axes) using the following formula [6,40]: $PlayerLoad™=[\sqrt{{\left(a_{y1}-a_{y-1}\right)}^{2}}+\surd {\left(a_{x1}-a_{x-1}\right)}^{2}+\surd {\left(a_{z1}-a_{z-1}\right)}^{2}]/100$, where fwd indicates movement in the anterior–posterior direction, side indicates movement in the medial–lateral direction, up indicates vertical movement, and t represents time, while EE was calculated as the number of inertial movements per minute (n·min) derived from the analysis of high- and medium-intensity inertial movements (accelerations, decelerations, and changes of direction).

To group the tasks based on the biomechanical load orientation, a two-step cluster analysis was conducted (average silhouette = 0.5) using the biomechanical parameters JUMPS, ACC, DEC, PL, and EE (Table 3). Exercises were grouped into tasks with low biomechanical load and tasks with high biomechanical load. A numerical value of 1 was assigned to low biomechanical load, while a value of 2 was assigned to high biomechanical load. To determine the biomechanical load of each task, an average was calculated for each task based on the numerical value of the cluster load ranging from 1 to 2.

### 2.4. Statistical Analysis

The mean, standard deviation (SD), and coefficient of variation (CV) were determined to describe the external physical load for each drill, while to describe the load orientation, the mean, median, and SD were utilized.

A Linear Mixed Model (LMM) was used to identify differences in external load and its orientation between drills (1v0-Individual Technical-Tactical-half court, 1v1 in longitudinal half court-28 × 7.5 m-, 2v0-Individual Technical-Tactical-half court, 2v2 full court, 3v0 full court, 3v3 full court, 3v3v3, 4v0 full court, 4v4 full court, 4v4v4, 5v0 full court, 5v5 full court, 5v5v5, and Eleven player break).

“Player” was used as a random effect. Tasks were included as nominal predictor variables in the LMM at 14 levels (1v0-Individual Technical-Tactical-half court, 1v1 full court in longitudinal half-28 × 7.5 m-, 2v0-Individual Technical-Tactical-half court, 2v2 full court, 3v0 full court, 3v3 full court, 3v3v3, 4v0 full court, 4v4 full court, 4v4v4, 5v0 full court, 5v5 full court, 5v5v5, Eleven player break).

Cohen’s effect size (ES) and the mean difference with 95% confidence intervals (CI) were determined for all pairwise comparisons and interpreted as follows: trivial = <0.20; small = 0.20–0.59; moderate = 0.60–1.19; large = 1.20–1.99; and very large = ≥2.00 [43]. All analyses were conducted using IBM SPSS for Windows (version 23, IBM Corporation, Armonk, NY, USA), except ES, which was calculated using a customized Microsoft Excel spreadsheet (version 16.0, Microsoft Corporation, Redmond, WA, USA).

## 3. Results

The descriptive analysis of each drill according to physical orientation (physiological or biomechanical) and specificity is presented in Table 4. The distribution of drills based on the orientation of the training load orientation is shown in Figure 1.

The descriptive analysis (mean ± SD, and % CV) of the external physical load of training drills and the effect size ± 95% CI of the differences between tasks (1v0-Individual Tactical-Technical-half-court vs. 1v1 full court in longitudinal middle-28 × 7.5 m-, 2v0-Individual Tactical-Technical-half-court, 2v2 full court, 3v0 full court, 3v3 full court, 3v3v3, 4v0 full court, 4v4 full court, 4v4v4, 5v0 full court, 5v5 full court, 5v5v5, Eleven Player Break) are shown in Figure 2.

Regarding the comparison for physiological load (Figure 3), it is notable that 4v4v4 was significantly lower than 3v3 full court (ES: −1.26), Eleven Player Fast Break (ES: −1.45), and 3v0 full court (ES: −1.30). Furthermore, Eleven Player Fast Break showed significantly higher values than 1v1 full court (ES: 1.23).

Regarding the comparison between tasks for the biomechanical load (Figure 3), 5v5 full court, 4v4 full court, and 3v3 full court showed significantly higher values than 4v0 full court (ES 5v5 full court: 1.37; ES 4v4 full court: 1.23; ES 3v3 full court: 1.49), 3v0 full court (ES 5v5 full court: 1.67; ES 4v4 full court: 1.85; ES 3v3 full court: 1.88), 2v0 half court (ES 5v5 full court: 1.71; ES 4v4 full court: 1.71; ES 3v3 full court: 1.38), and 1v0 half court (ES 5v5 full court: 1.59; ES 4v4 full court: 1.58; ES 3v3 full court: 1.31). Additionally, 4v4 full court also showed significantly higher values than 5v0 full court (ES: 1.23).

Concerning 3v3v3, the results showed significantly higher values than 3v0 full court (ES: 1.65), 2v0 half court (ES: 1.38), and 1v0 half court (ES: 1.31). Moreover, Eleven Player Fast Break reached significantly higher values than 3v0 full court (ES: 1.29). Regarding 1v1 full court, it showed significantly higher values compared to 5v0 full court (ES: 1.23), 4v0 full court (ES: 1.56), 3v0 full court (ES: 1.93), 2v0 half court (ES: 1.78), and 1v0 half court (ES: 1.64). Finally, 5v0 full court obtained significantly higher values than 4v0 full court (ES: 1.30).

Regarding the comparisons of physiological and biomechanical load, significant differences (*p* < 0.05) with effect sizes ranging from trivial to very large are shown in Table 5.

## 4. Discussion

The aim of this article was twofold: firstly, to know and describe the physiological and biomechanical loads of the different tasks used in basketball training and, subsequently, to make a practical proposal of these tasks throughout a typical training week.

In relation to the first goal, the present study has allowed us to categorize the different tasks used in basketball training under the perspective of physiological load or biomechanical load. One of the main reasons for conducting this study is that, as reflected in several specific investigations [2,5], there are still many limitations in the research carried out to date on this topic given the large number of variables that can modify the load imposed by each of the tasks used in basketball. Moreover, this aspect is usually analyzed under the view of high or low load, i.e., under the perspective of the amount of load, but not under the perspective of the nature of the training load, which can be physiological or neuromuscular in nature [23]. For example, in the results of the review by O’Grady et al. (2020) [2], it is pointed out that the results of different studies analyzed [29,44] suggest that SSGs with fewer players (2v2, 3v3) cause a greater training load, both internally and externally, compared to SSGs with a greater number of players (4v4, 5v5), and that exercises used in full court also involve a greater external load than those performed in half court, regardless of team size. Similarly, Clemente (2016) [45] suggests that involving fewer players in SSGs means higher intensity compared to 5v5. Atli et al. (2013) [46] also suggest that when the number of players remains constant but the playing area increases (leading to an increase in the relative distance to be covered), significant differences arise in the load of each of the SSGs. While most of the results found so far are in line with these ideas [29,30,44], they are still very general, because as the results of the present research show, under the perspective of biomechanical and physiological loading, these results can be nuanced, and therefore, would be a better help for coaches when designing training sessions.

Therefore, the results of this study are considered relevant, as it is the first research, to the best of the researchers’ knowledge, to classify the different training tasks based on the nature of the load, i.e., physiological load or biomechanical load. The main results obtained can be seen graphically in Figure 1. In summary, it could be said that those tasks that cover more space (full court vs. half court) and with fewer defenders (3vs3, 2vs2, 11 counterattack, 5v0, 4v0, and 3v0) have a higher physiological load, while tasks without defense tend to have lower values of biomechanical load. However, those tasks with less space and more defenders (3v3v3, 4v4v4, 5v5, and 4v4) have a higher biomechanical load.

It could be concluded that the higher biomechanical load is closely related to the presence of defenders. However, in the case of 1v0 and 2v0 tasks, although less demanding, it should be noted that they present a certain biomechanical load (as they are normally linked to technical work and, therefore, accumulate a high number of jumps/min). In this sense, the study by Schelling & Torres. (2016) [29] also found that, for variables such as accelerations per minute, half-court exercises were more demanding. Specifically, 2v2 and 5v5 in half court showed the highest values for accelerations per minute among the different SSGs analyzed. In the study by Olthof et al. (2021) [24], they found a positive association between the biomechanical load of the training sessions with the players’ statistics during the match and suggested that biomechanical loads were good predictors for game performance, in the way that excessive biomechanical loads from training may negatively impact game performance. Finally, Castillo et al. (2021) [47] found significant differences in high decelerations and jumps when considering the interaction of the defensive style factors and the outcome of game-based drills.

Although numerous modulators of the external load (opposition/non-opposition, number of opponents, type of opposition, limitation of technical actions, or feedback from the coach) have been described in the specific literature, the playing space seems to be the fundamental variable in the regulation of the intensity of the exercises (i.e., [2,4,45]). The m^2^/player ratio determines and guides the task load. By modifying/restricting the absolute (total m^2^) or relative (m^2^/player) spaces, the biomechanical and physiological demands of the exercises can be modulated to a large extent. However, the results obtained in the present study qualify this idea, as it is not only space that will be the main modulator of the load experienced by the player, but also the presence or absence of defenders. Therefore, the combination of these two variables will be the main modulator of tasks to impose a greater physiological or biomechanical load on the athlete. This could coincide with the results obtained by Sansone et al. (2023) [4], who, while analyzing the training tasks used in basketball from a different perspective, come to the conclusion that the modification of the number of players involved in the task and the space available to the player should be used to modify the external load experienced.

However, in the present study we should avoid having a dichotomous view of this perspective, and it would be much more convenient to understand this analysis not as an analysis of the training tasks between two extremes (high physiological load or high biomechanical load), but as a continuum between these possibilities. In this sense, according to the results obtained, it would be much more advisable to classify the tasks as exercises fundamentally of biomechanical orientation (1v1 full court, 3v3v3, 4v4v4v, and 5v5v5), exercises fundamentally of physiological orientation (3v0 full court, 4v0 full court, and 5v5v5), low-intensity mixed-orientation drills (1v0 half court and 2v0 half court), and high-intensity mixed-orientation drills (official match, 4v4 full court, 3v3 full court, 5v5 full court, counter attack of 11, and 2v2 full court).

In relation to the second goal, it is necessary to highlight its clear practical application, as the present analysis allows us to model training based on the knowledge of the real impact of each task. With this objective we could define different types of sessions as shown in Table 6. Based on the results obtained, depending on the objectives we are looking for when designing the training, we will be able to select more suitable tasks for each of these objectives:

Another main application of this task classification is the weekly design of training according to the number of competitions and their location, as it appears in Figure 4. Weekly tapering or short-term tapering is the weekly adjustment of the training load with the objective of obtaining an optimal performance for the competition. This programming involves an overload phase (the day’s farthest away from the competition) and a tapering phase (the days closest to the competition). Since there is a high sensitivity of physical qualities to tapering in team sports, understanding the differences in the demands of the different tasks allows us to improve the exercise selection system and training design, especially when seeking to optimize weekly performance. It should be noted that there are studies that have revealed a large interindividual variability in individual sports in response to tapering.

Greater control of the training stimulus and the adaptations that occur during periods of progressive loading and tapering, especially during periods of intense physiological and psychological stress, that is, prior to competition, could improve the training design and management of the training load. Therefore, the choice of exercises could be crucial to establish an optimal pre-competition physical load.

The main limitation of the present study is that probably not all of the possible tasks to be used in basketball training have been analyzed, although it can be observed that most of the tasks normally used in training sessions are included. However, what is relevant is that it allows for a more correct design of the training sessions by placing the tasks with a greater biomechanical impact at the beginning of the sessions and trying to place the tasks with a greater physiological orientation towards the end of the training. Similarly, if the objective of the planned training is more related to fatigue endurance work, the predominant tasks should be those with a high physiological load, whereas if the objective of the training is mainly tactical or strategic, the tasks to be used will have a high biomechanical load. Additionally, one of the main limitations of this study is the small and highly specific sample size, which restricts the generalizability of the findings. In this regard, the results are closely related to the specific coaching methodology and training design employed in this study, limiting their applicability to other coaching approaches or contexts.

Future research should aim to include larger and more diverse samples to enhance the generalizability of the findings. Expanding the study to include players from various competitive levels, age groups, and geographical regions would provide a better understanding of the physiological and biomechanical demands in basketball training. Furthermore, exploring how different coaching methodologies impact these variables could offer valuable insights for practitioners seeking to adapt the findings to their specific needs and training environments.

## 5. Conclusions

This study classifies basketball training tasks according to their physiological or biomechanical load, showing that tasks with more space and fewer defenders impose higher physiological loads, while those with less space and more defenders increase the biomechanical load. For training design, it is recommended to place tasks with higher biomechanical load at the beginning of the session and those with physiological orientation toward the end. Manipulating space and the presence of defenders allows for adjusting task intensity to meet specific objectives, optimizing performance and avoiding overtraining.

## Figures and Tables

**Figure 1 sensors-25-00262-f001:**
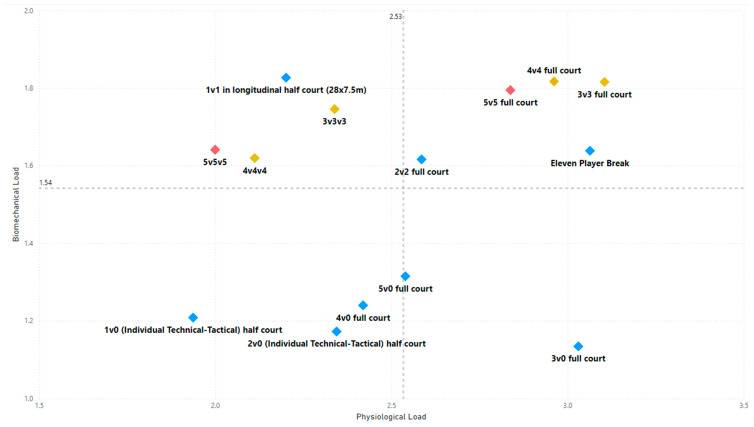
Distribution of drills based physical orientation (physiological or biomechanical) and specificity. *Note:* The colors are related to the specificity of the exercises, as shown in Table 4, evolving from less specific exercises -blue- to more specific exercises -red-.

**Figure 2 sensors-25-00262-f002:**
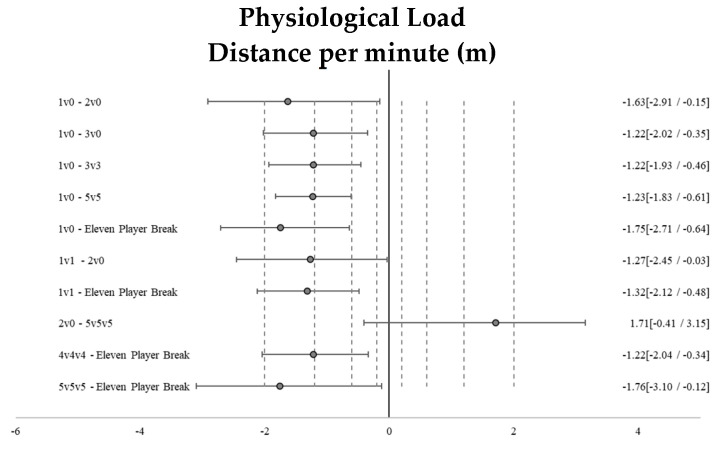
Standardized differences (Cohen’s d) and their respective 95% confidence intervals (CI) between the training tasks that showed significant large–very large size differences for physiological and biomechanical loads.

**Figure 3 sensors-25-00262-f003:**
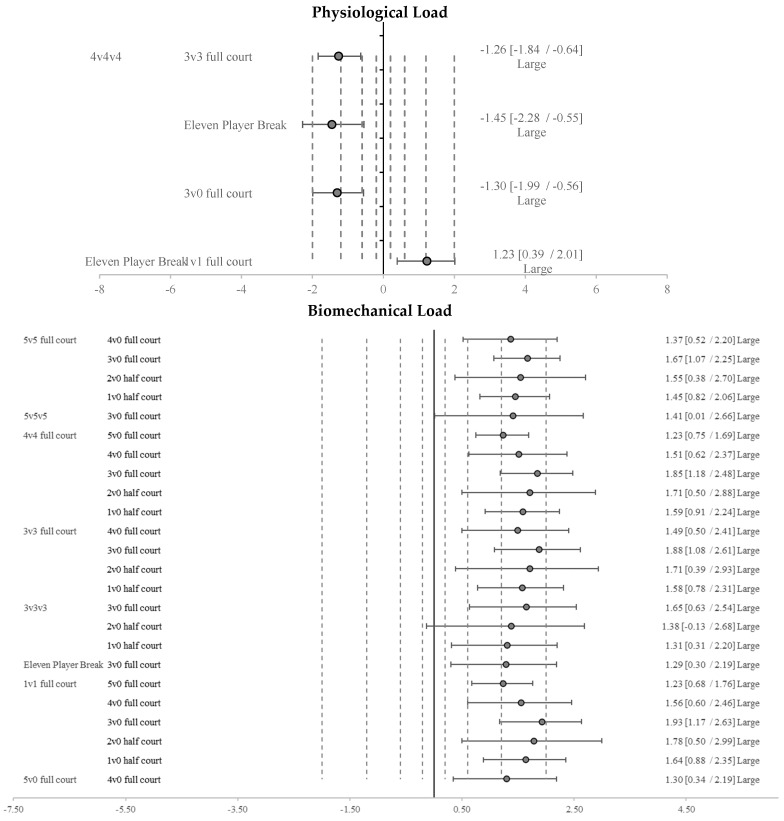
Standardized differences (Cohen’s d) and their respective 95% confidence intervals (CI) between training tasks and match tasks for physiological and biomechanical load. Notes: The dashed line represents the magnitude of the effect from large to very large.

**Figure 4 sensors-25-00262-f004:**
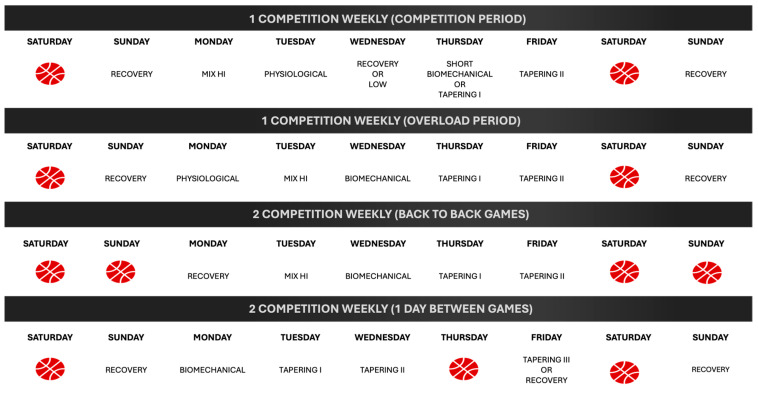
Weekly training design according to the number of competitions. Notes: The basketball ball represents an official game according to a typical calendar in basketball leagues.

**Table 1 sensors-25-00262-t001:** Drill classification based on their specificity from 5 to 0 (Schelling & Torres, 2016).

4	5v5 full court	A 5v5 game is played with 10 players on the court at the same time. The number of consecutive plays and the work-to-rest ratio vary depending on the coach’s feedback and the pauses they implement.	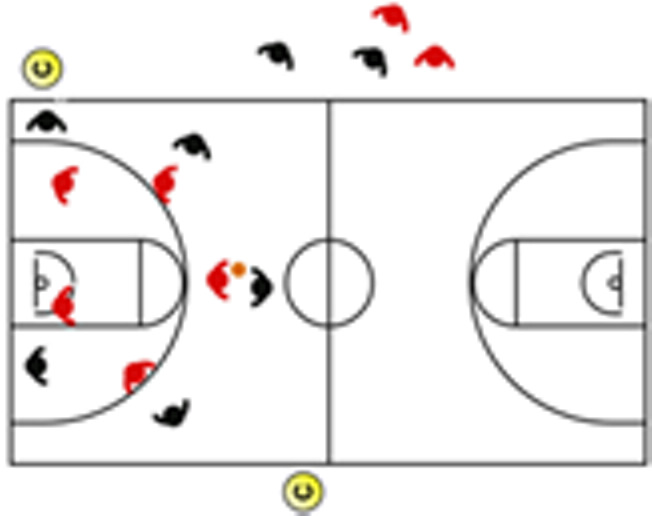
4	5v5v5	A 5v5v5 game is played where the defending team transitions to offense and attacks the opposite basket.	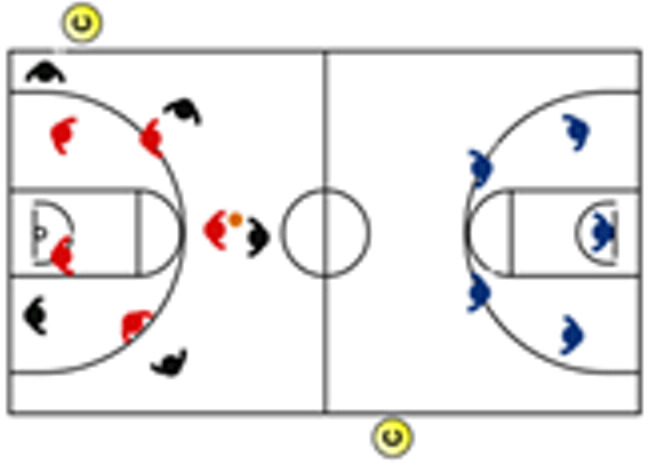
4	4v4 full court	A 4v4 game is played with 8 players on the court at the same time. When the offensive play ends, the defending team transitions to offense and attacks the same team at the opposite basket. The number of consecutive plays and the work-to-rest ratio ranges between 3 and 6, depending on the coach’s feedback and pauses	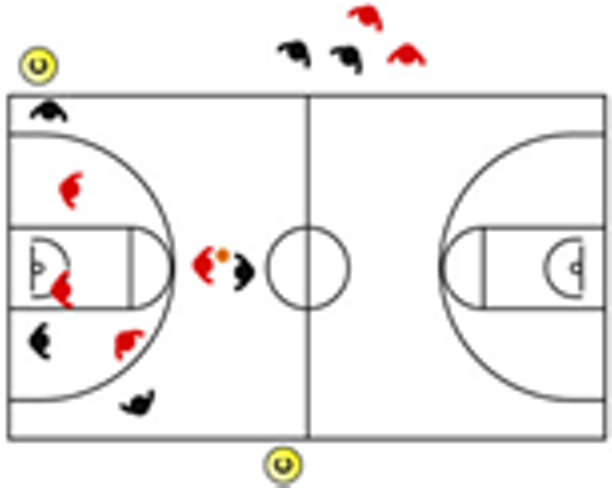
4	4v4v4	A 4v4v4 game is played where the defending team transitions to offense and attacks the opposite basket, where another team is waiting to defend.	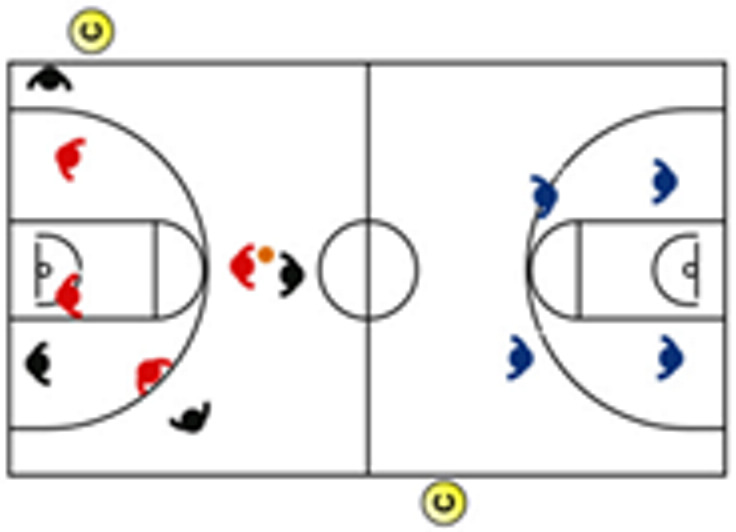
3	3v3 Full court	A 3v3 game is played with 6 players on the court at the same time. When the offensive play ends, the defending team transitions to offense and attacks the same team at the opposite basket. The number of consecutive plays and the work-to-rest ratio vary between 3 and 6, depending on the coach’s feedback and pauses.	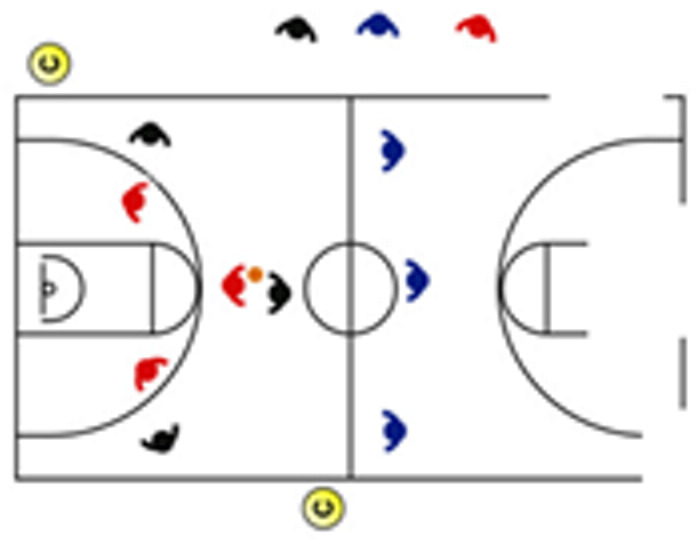
3	3v3v3	A 3v3v3 game is played where the defending team transitions to offense and attacks the opposite basket, where another team is waiting to defend.	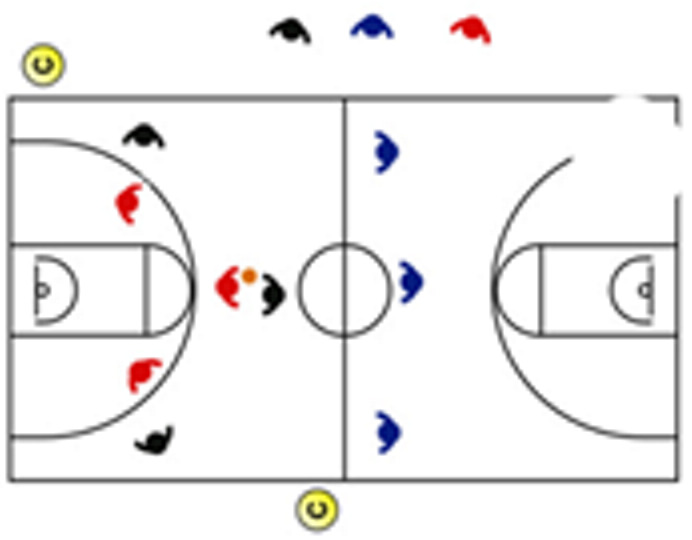
3	Eleven Player Break	A continuous 3v2 situation is played. Among the five players involved, the one who gains possession when the play ends (whether through a basket, rebound, or turnover) attacks on the opposite side with two players positioned in the corners against two defenders waiting on the other side.	** 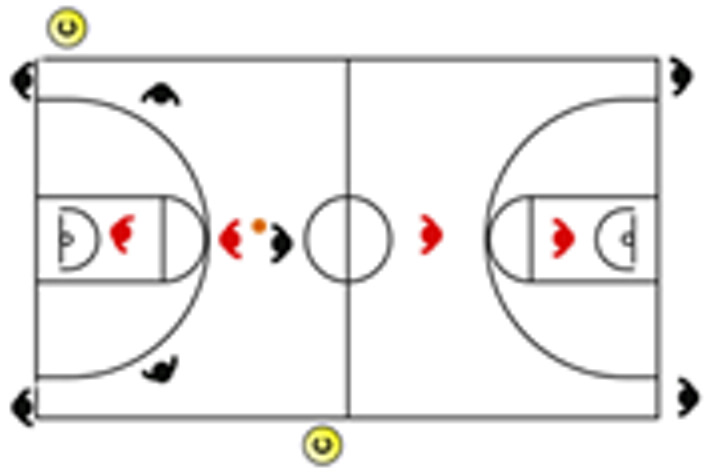 **
3	2v2 full court	A 2v2 game is played where, after an offensive play, the team defends at the opposite basket. Following the defensive effort, the team passes to one of the two teammates positioned to transition and attack on the opposite court.	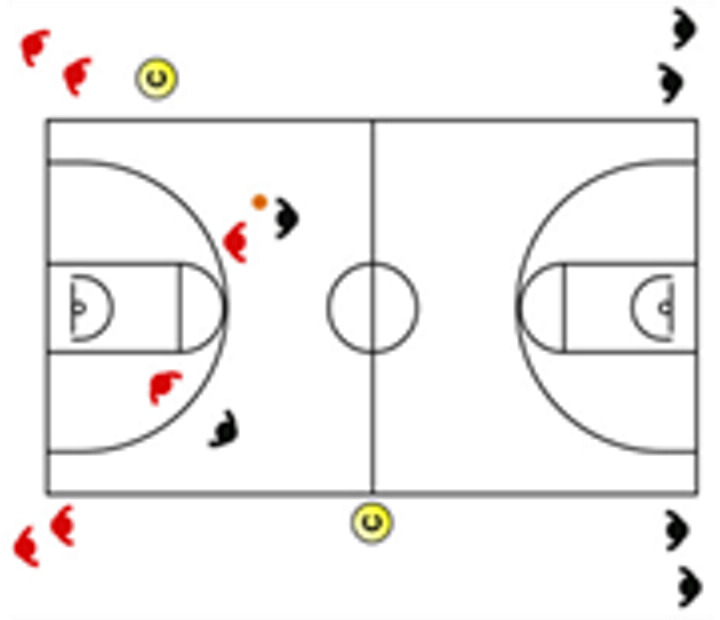
2	1v1 in longitudinal half court (28 × 7.5 m)	The attacking player must attempt to drive past and score after playing a 1v1. Once the offensive play is over, the player who attacked transitions to defense.	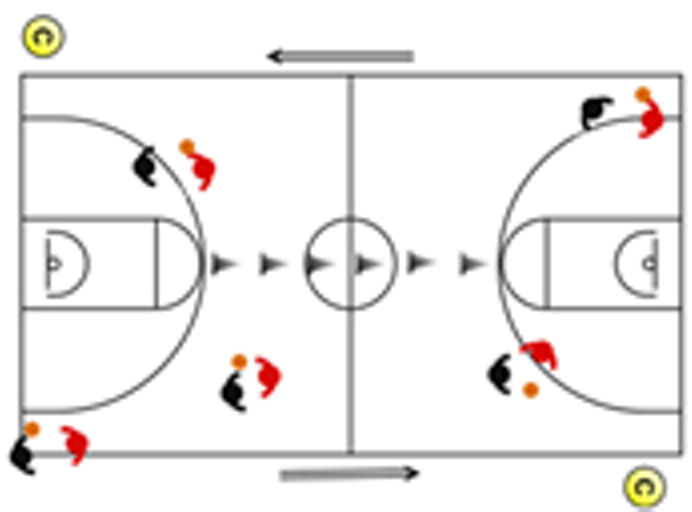
2	5v0 full court	A 5v0 drill is performed at midcourt, followed by another drill at the opposite end. After completing these drills, 5 new players enter the court.	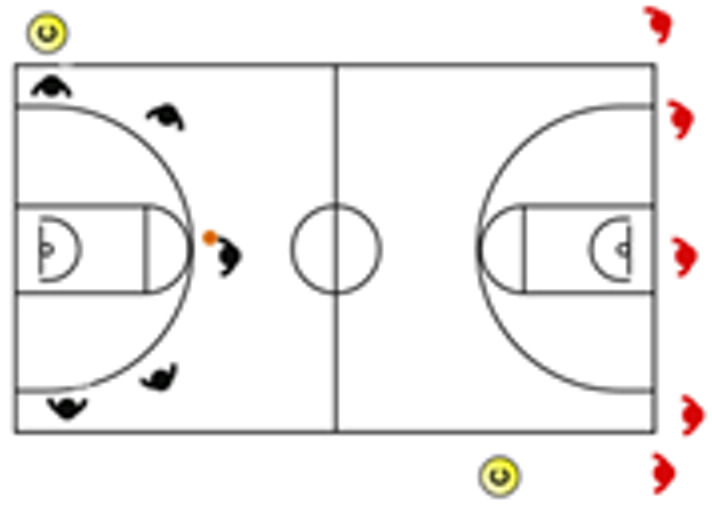
2	4v0 full court	A 4v0 drill is performed at midcourt, followed by another drill at the opposite end. After completing these drills, 4 new players enter the court.	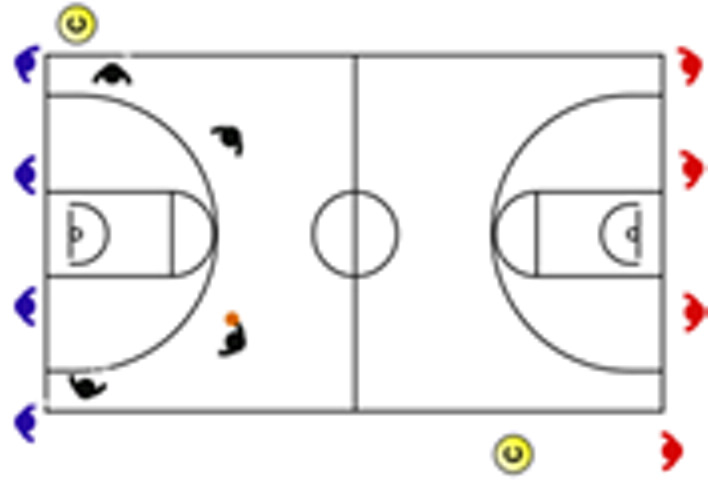
2	3v0 full court	A 3v0 drill is performed at midcourt, followed by another drill at the opposite end. After completing these drills, 3 new players enter the court.	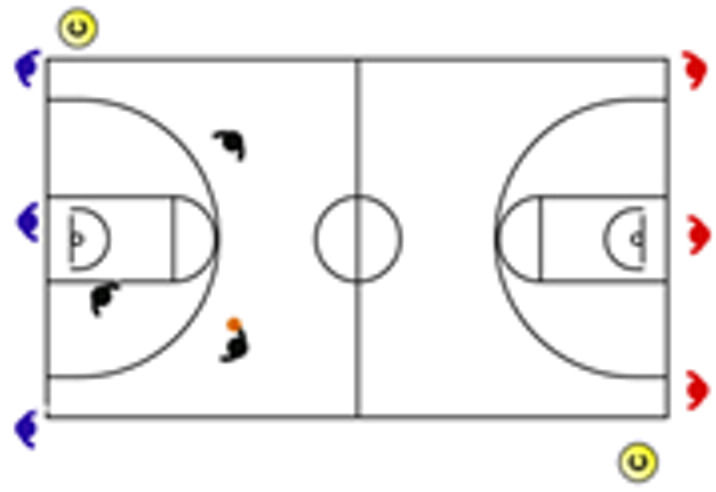
2	2v0 (Individual Technical-Tactical) half court	Different individual technical-tactical situations are practiced without opposition in a 2v0 setting	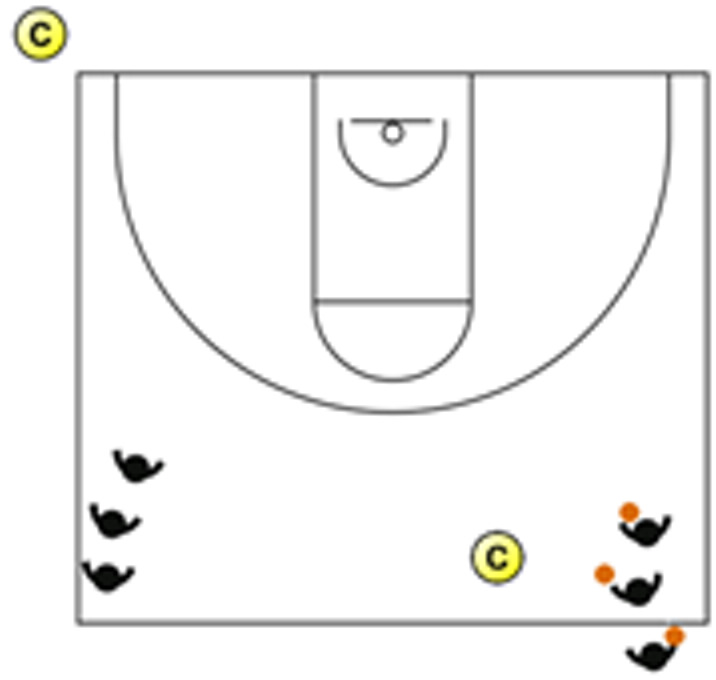
2	1v0 (Individual Technical-Tactical) half court	Different individual technical-tactical situations are practiced without opposition in a 1v0 setting.	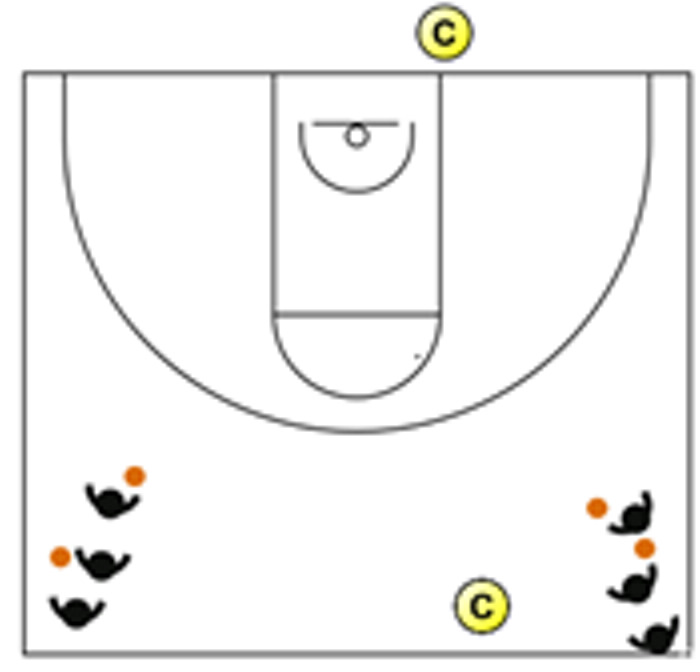

**Table 2 sensors-25-00262-t002:** Cluster analysis identifying drill groups based on physiological load parameters.

Variables	Physiological Load			
	Low	Medium	High	Very High
Distance per minute (m)	18.56	62.09	75.28	80.50
Standing–walking (<7 km·h^−1^)	15.53	31.75	65.40	34.35
Jogging (7–14 km·h^−1^)	3.13	21.36	51.81	28.05
Running (14.01–18 km·h^−1^)	0.85	6.37	17.32	12.29
High-speed running (>18 km·h^−1^)	0.22	2.42	3.77	6.02
Sample size (N)	141	1831	122	1042
Sample proportion (%)	4.5%	58.4%	3.9%	32.2%
Bayesian information criterion (BIC)	9214.44			
Average silhouette	0.5			

*Note*: The value of each physiological load variable is presented as the mean and standard deviation for each group, and the sample size indicates the number of tasks included in each group.

**Table 3 sensors-25-00262-t003:** Cluster analysis identifying drill groups based on biomechanical load parameters.

Variables	Biomechanical Load	
	Low	High
Accelerations per minute	1.35	2.71
Decelerations per minute	1.20	3.38
Explosive efforts per minute	1.56	3.26
PlayerLoad per minute	5.91	8.42
Jumps per minute	0.65	0.73
Sample size (N)	128	2124
Sample proportion (%)	47.4%	67.7%
Bayesian Information Criterion (BIC)	10,677.49	
Average silhouette	0.5	

*Notes:* The value of each physiological load variable is presented as the mean and standard deviation of each group, and the sample size indicates the number of tasks included in each group.

**Table 4 sensors-25-00262-t004:** Descriptive statistics for each drill according to physical orientation (physiological or biomechanical) and specificity.

Drill	Specificity	Physiological Load		Biomechanical Load	
		Median	Mean ± SD (% CV)	Median	Mean ± SD (% CV)
**5v5 full court**	**4**	2	2.84 ± 0.99 (35%)	2	1.79 ± 0.40 (22%)
**5v5v5**		2	2.00 ± 0.00 (0%)	2	1.64 ± 0.48 (29%)
**4v4 full court**	**3**	2	2.96 ± 1.01 (34%)	2	1.82 ± 0.38 (21%)
**4v4v4**		2	2.11 ± 0.51 (24%)	2	1.62 ± 0.48 (30%)
**3v3 Full court**		4	3.11 ± 0.99 (32%)	2	1.82 ± 0.38 (21%)
**3v3v3**		2	2.34 ± 0.75 (32%)	2	1.75 ± 0.43 (25%)
**Eleven Player Break**	**2**	4	3.06 ± 1.00 (33%)	2	1.64 ± 0.48 (29%)
**2v2 full court**		2	2.59 ± 0.99 (38%)	2	1.62 ± 0.48 (30%)
**1v1 in longitudinal half court (28 × 7.5 m)**		2	2.20 ± 0.62 (28%)	2	1.83 ± 0.37 (20%)
**5v0 full court**		2	2.54 ± 0.92 (36%)	1	1.32 ± 0.46 (35%)
**4v0 full court**		2	2.42 ± 0.81 (33%)	1	1.24 ± 0.43 (35%)
**3v0 full court**		3	3.03 ± 0.97 (32%)	1	1.13 ± 0.34 (30%)
**2v0 (Individual Technical-Tactical) half court**		2	2.34 ± 0.72 (31%)	1	1.17 ± 0.38 (32%)
**1v0 (Individual Technical-Tactical) half court**		2	1.94 ± 0.24 (12%)	1	1.21 ± 0.40 (33%)

**Table 5 sensors-25-00262-t005:** Comparisons for each drill according to physiological or biomechanical orientation.

		Physiological Load		Biomechanical Load	
		Dif. Mean [I/S]	Sig.	Dif. Mean [I/S]	Sig.
**1v0**	*1v1*	−0.26 [−0.63/0.10]	1.000	−0.62 * [−0.79/−0.45]	0.000
	*2v0*	−0.41 [−1.06/0.25]	1.000	0.04 [−0.26/0.34]	1.000
	*2v2*	−0.64 * [−1.09/−0.21]	0.000	−0.41 * [−0.61/−0.2]	0.000
	*3v0*	−1.09 * [−1.54/−0.65]	0.000	0.07 [−0.13/0.28]	1.000
	*3v3*	−1.16 * [−1.54/−0.80]	0.000	−0.61 * [−0.78/−0.44]	0.000
	*3v3v3*	−0.40 [−0.91/0.11]	0.670	−0.54 * [−0.77/−0.3]	0.000
	*4v0*	−0.48 [−1.02/0.06]	0.190	−0.03 [−0.28/0.22]	1.000
	*4v4*	−1.02 * [−1.37/−0.68]	0.000	−0.61 * [−0.77/−0.45]	0.000
	*4v4v4*	−0.17 [−0.56/0.21]	1.000	−0.41 * [−0.59/−0.24]	0.000
	*5v0*	−0.60 * [−0.99/−0.22]	0.000	−0.11 [−0.28/0.07]	1.000
	*5v5*	−0.90 * [−1.23/−0.57]	0.000	−0.59 * [−0.74/−0.43]	0.000
	*5v5v5*	−0.06 [−0.65/0.52]	1.000	−0.43 * [−0.7/−0.16]	0.000
	*Eleven Player Break*	−1.13 * [−1.68/−0.58]	0.000	−0.43 * [−0.68/−0.18]	0.000
**1v1**	*2v0*	−0.14 [−0.75/0.46]	1.000	0.65 * [0.38/0.93]	0.000
	*2v2*	−0.38 * [−0.75/−0.02]	0.020	0.21 * [0.05/0.38]	0.000
	*3v0*	−0.83 * [−1.19/−0.47]	0.000	0.69 * [0.53/0.86]	0.000
	*3v3*	−0.90 * [−1.17/−0.64]	0.000	0.01 [−0.11/0.13]	1.000
	*3v3v3*	−0.14 [−0.58/0.31]	1.000	0.08 [−0.12/0.28]	1.000
	*4v0*	−0.22 [−0.69/0.26]	1.000	0.59 * [0.37/0.8]	0.000
	*4v4*	−0.76 * [−0.99/−0.53]	0.000	0.01 [−0.09/0.11]	1.000
	*4v4v4*	0.09 [−0.19/0.37]	1.000	0.21 * [0.08/0.34]	0.000
	*5v0*	−0.34 * [−0.62/−0.05]	0.000	0.51 * [0.38/0.64]	0.000
	*5v5*	−0.64 * [−0.85/−0.43]	0.000	0.03 [−0.06/0.13]	1.000
	*5v5v5*	0.20 [−0.33/0.73]	1.000	0.19 [−0.06/0.43]	0.800
	*Eleven Player Break*	−0.86 * [−1.35/−0.38]	0.000	0.19 [−0.03/0.41]	0.340
**2v0**	*2v2*	−0.24 [−0.89/0.41]	1.000	−0.44 * [−0.74/−0.14]	0.000
	*3v0*	−0.69 * [−1.34/−0.03]	0.030	0.04 [−0.26/0.34]	1.000
	*3v3*	−0.76 * [−1.37/−0.16]	0.000	−0.64 * [−0.92/−0.37]	0.000
	*3v3v3*	0.01 [−0.70/0.71]	1.000	−0.57 * [−0.89/−0.25]	0.000
	*4v0*	−0.08 [−0.80/0.65]	1.000	−0.07 [−0.4/0.26]	1.000
	*4v4*	−0.62 * [−1.21/−0.03]	0.030	−0.65 * [−0.91/−0.37]	0.000
	*4v4v4*	0.23 [−0.38/0.85]	1.000	−0.45 * [−0.73/−0.17]	0.000
	*5v0*	−0.20 [−0.81/0.42]	1.000	−0.14 [−0.42/0.14]	1.000
	*5v5*	−0.49 [−1.08/0.09]	0.340	−0.62 * [−0.89/−0.36]	0.000
	*5v5v5*	0.34 [−0.41/1.10]	1.000	−0.47 * [−0.82/−0.12]	0.000
	*Eleven Player Break*	−0.72 [−1.45/0.01]	0.060	−0.47 * [−0.8/−0.13]	0.000
**2v2**	*3v0*	−0.45 * [−0.89/0.00]	0.050	0.48 * [0.28/0.68]	0.000
	*3v3*	−0.52 * [−0.88/−0.16]	0.000	−0.20 * [−0.37/−0.03]	0.000
	*3v3v3*	0.25 [−0.26/0.76]	1.000	−0.13 [−0.36/0.1]	1.000
	*4v0*	0.17 [−0.37/0.70]	1.000	0.38 * [0.13/0.62]	0.000
	*4v4*	−0.38 * [−0.71/−0.04]	0.010	−0.20 * [−0.36/−0.05]	0.000
	*4v4v4*	0.47 * [0.10/0.85]	0.000	0 [−0.18/0.17]	1.000
	*5v0*	0.05 [−0.33/0.43]	1.000	0.30 * [0.13/0.48]	0.000
	*5v5*	−0.25 [−0.58/0.08]	0.790	−0.18 * [−0.33/−0.03]	0.000
	*5v5v5*	0.59 * [0.00/1.17]	0.050	−0.02 [−0.29/0.24]	1.000
	*Eleven Player Break*	−0.48 [−1.03/0.07]	0.250	−0.02 [−0.27/0.23]	1.000
**3v0**	*3v3*	−0.07 [−0.44/0.29]	1.000	−0.68 * [−0.85/−0.51]	0.000
	*3v3v3*	0.69 * [0.18/1.20]	0.000	−0.61 * [−0.85/−0.38]	0.000
	*4v0*	0.61 * [0.07/1.15]	0.010	−0.11 [−0.35/0.14]	1.000
	*4v4*	0.07 [−0.27/0.41]	1.000	−0.68 * [−0.84/−0.53]	0.000
	*4v4v4*	0.92 * [0.54/1.30]	0.000	−0.49 * [−0.66/−0.31]	0.000
	*5v0*	0.49 * [0.11/0.87]	0.000	−0.18 * [−0.36/−0.01]	0.030
	*5v5*	0.19 [−0.14/0.52]	1.000	−0.66 * [−0.81/−0.51]	0.000
	*5v5v5*	1.03 * [0.44/1.62]	0.000	−0.51 * [−0.78/−0.24]	0.000
	*Eleven Player Break*	−0.03 [−0.58/0.52]	1.000	−0.50 * [−0.76/−0.25]	0.000
**3v3**	*3v3v3*	0.77 * [0.32/1.21]	0.000	0.07 [−0.13/0.27]	1.000
	*4v0*	0.69 * [0.21/1.16]	0.000	0.58 * [0.36/0.79]	0.000
	*4v4*	0.14 [−0.09/0.38]	1.000	0 [−0.11/0.11]	1.000
	*4v4v4*	0.99 * [0.71/1.28]	0.000	0.20 * [0.06/0.33]	0.000
	*5v0*	0.57 * [0.28/0.85]	0.000	0.50 * [0.37/0.63]	0.000
	*5v5*	0.27 * [0.05/0.48]	0.000	0.02 [−0.08/0.12]	1.000
	*5v5v5*	1.11 * [0.57/1.64]	0.000	0.17 [−0.07/0.42]	1.000
	*Eleven Player Break*	0.04 [−0.45/0.53]	1.000	0.18 [−0.05/0.4]	0.620
**3v3v3**	*4v0*	−0.08 [−0.68/0.51]	1.000	0.51 * [0.23/0.78]	0.000
	*4v4*	−0.62 * [−1.05/−0.20]	0.000	−0.07 [−0.27/0.12]	1.000
	*4v4v4*	0.23 [−0.23/0.68]	1.000	0.13 [−0.08/0.34]	1.000
	*5v0*	−0.20 [−0.66/0.26]	1.000	0.43 * [0.22/0.64]	0.000
	*5v5*	−0.50 * [−0.91/−0.08]	0.000	−0.05 [−0.24/0.14]	1.000
	*5v5v5*	0.34 [−0.30/0.98]	1.000	0.1 [−0.19/0.4]	1.000
	*Eleven Player Break*	−0.73 * [−1.33/−0.12]	0.000	0.11 [−0.17/0.38]	1.000
**4v0**	*4v4*	−0.54 * [−1.00/−0.08]	0.000	−0.58 * [−0.79/−0.37]	0.000
	*4v4v4*	0.31 [−0.18/0.80]	1.000	−0.38 * [−0.6/−0.16]	0.000
	*5v0*	−0.12 [−0.61/0.37]	1.000	−0.08 [−0.3/0.15]	1.000
	*5v5*	−0.42 [−0.87/0.03]	0.120	−0.56 * [−0.76/−0.35]	0.000
	*5v5v5*	0.42 [−0.24/1.08]	1.000	−0.40 * [−0.7/−0.1]	0.000
	*Eleven Player Break*	−0.64 * [−1.27/−0.02]	0.040	−0.40 * [−0.69/−0.11]	0.000
**4v4**	*4v4v4*	0.85 * [0.59/1.10]	0.000	0.20 * [0.08/0.31]	0.000
	*5v0*	0.42 * [0.16/0.68]	0.000	0.50 * [0.38/0.62]	0.000
	*5v5*	0.12 [−0.05/0.29]	1.000	0.02 [−0.06/0.1]	1.000
	*5v5v5*	0.96 * [0.45/1.48]	0.000	0.18 [−0.06/0.41]	0.980
	*Eleven Player Break*	−0.10 [−0.57/0.37]	1.000	0.18 [−0.04/0.39]	0.410
**4v4v4**	*5v0*	−0.43 * [−0.74/−0.12]	0.000	0.31 * [0.16/0.45]	0.000
	*5v5*	−0.73 * [−0.97/−0.49]	0.000	−0.18 * [−0.28/−0.07]	0.000
	*5v5v5*	0.11 [−0.43/0.65]	1.000	−0.02 [−0.27/0.23]	1.000
	*Eleven Player Break*	−0.95 * [−1.45/−0.45]	0.000	−0.02 [−0.25/0.21]	1.000
**5v0**	*5v5*	−0.30 * [−0.54/−0.06]	0.000	−0.48 * [−0.59/−0.37]	0.000
	*5v5v5*	0.54 [0.00/1.08]	0.050	−0.33 * [−0.57/−0.08]	0.000
	*Eleven Player Break*	−0.52 * [−1.03/−0.02]	0.030	−0.32 * [−0.55/−0.09]	0.000
	*Partido oficial*	−0.66 * [−1.00/−0.32]	0.000	−0.59 * [−0.75/−0.44]	0.000
	*Tiros libres*	1.54 * [1.16/1.92]	0.000	0.32 * [0.14/0.49]	0.000
**5v5**	*5v5v5*	0.84 * [0.33/1.34]	0.000	0.15 [−0.08/0.39]	1.000
	*Eleven Player Break*	−0.23 [−0.69/0.24]	1.000	0.16 [−0.06/0.37]	1.000
**5v5v5**	*Eleven Player Break*	−1.06 * [−1.73/−0.39]	0.000	0 [−0.3/0.31]	1.000

Notes: The * means statistically significant differences.

**Table 6 sensors-25-00262-t006:** Different types of sessions according to the objectives and physical orientation (physiological or biomechanical).

Orientation	Session Duration	Tasks		Task Duration
		Main:	3v0-4v0-5v0	15–20 min
Physiological	60–90 min	Reinforcing:	3v3-4v4-5v5	10–12 min
		Accessories:	1v0-2v0	10–12 min
		Main:	1v1FC-3v3v3-4v4v4-5v5v5	15–20 min
Biomechanical	60–90 min	Reinforcing:	5v5HC	10–12 min
		Accessories:	1v0-2v0	10–12 min
		Main:	2v2 FC-11PB-3v3-4v4-5v5-SGs	15–20 min
Mixed high intensity	60–90 min	Reinforcing:	-	-
		Accessories:	1v0-2v0	10–12 min
		Main:	3v3v3-4v4v4-5v5v5-4v4-5v5	15–20 min
Tapering I	60–75 min	Reinforcing:	3v3	10–12 min
		Accessories:	1v0-2v0	10–12 min
		Main:	5v5v5-4v4-5v5	15–20 min
Tapering II	45–60 min	Reinforcing:	5v0	10–12 min
		Accessories:	1v0-2v0	10–12 min
		Main:	5v0	8–10 min
Tapering III	30–45 min	Reinforcing:	5v5v5-5v5 (limited contact, no tape)	5–8 min
		Accessories:	1v0-2v0	10–12 min

## Data Availability

The data used are the property of the basketball team in which they were recorded and are not publicly available. The authors do not have permission to share the data publicly. However, should anyone be interested in learning more, please contact the authors directly.

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
