# Peer review of "Description and Classification of Training Drills, Based on Biomechanical and Physiological Load, in Elite Basketball"

_sensors, 2025, doi:10.3390/s25010262_

Round 1

Reviewer 1 Report

Comments and Suggestions for Authors

Generally speaking, the paper is interesting and well-written. 

One comment is that more samples (e.g., 100 basketball players had their external load variables monitored across 1000 training sessions) are suggested to be used for more accurate and more convincing results. 

Author Response

Dear reviewer. 

Thank you for all your comments. Please, see the attachment.

Kind Regards.

Reviewer 2 Report

Comments and Suggestions for Authors

See attached

Author Response

(The authors gave the same response as above.)

Reviewer 3 Report

Comments and Suggestions for Authors

This paper describes the physiological and biomechanical demands of various tasks used in basketball training and, subsequently, provides a practical application of these tasks across a typical training week. The manuscript has certain innovation and application value, but there are some issues that need to be addressed.

1. The title of the paper seems not to meet the basic norms. Please revise it if possible.

2. The content of the literature review is not sufficient. Provide a detailed introduction and discussion of the current methods related to the methods in this paper, and cite more literature, which may not be related to basketball. Meanwhile, 32 references are cited in the first two paragraphs of the literature, which is incredible. Please pay attention to streamlining and citing more relevant literature, such as some literature related to basketball training, such as Shooting Prediction Based on Vision Sensors and ANN - Enhanced IoT Wristband for Basketball Shooting Motion Analysis.

3. Please add a discussion on the contributions of this paper in the introduction section.

4. The analysis in this paper mainly relies on biological and physical variables, so physical health conditions may greatly affect the experimental results. Please add relevant content regarding the health status of the participants.

5. The authors carried out cluster analysis based on the measured variables in the research. How were the criteria divided in this process? Was it completed through clustering algorithms or based on expert opinions?

6. The content of the summary section is too little and needs to be enhanced. Please add content related to the limitations of this paper and the next - step work.

7. There are some grammatical errors in the paper. Please pay attention to modifying them.

Comments on the Quality of English Language

There are some grammatical errors in the paper. Please pay attention to modifying them.

Author Response

(The authors gave the same response as above.)

Round 2

Reviewer 3 Report

Comments and Suggestions for Authors

This manuscript may be accepted in its current version

Comments on the Quality of English Language

The English could be improved to more clearly express the research.